# TDP-43 Regulates Rab4 Levels to Support Synaptic Vesicle Recycling and Neuromuscular Connectivity in *Drosophila* and Human ALS Models

**DOI:** 10.3390/ijms262211030

**Published:** 2025-11-14

**Authors:** Monsurat Gbadamosi, Giulia Romano, Michela Simbula, Giulia Canarutto, Linda Ottoboni, Stefania Corti, Fabian Feiguin

**Affiliations:** 1International Centre for Genetic Engineering and Biotechnology, Padriciano 99, 34149 Trieste, Italygiulia.romano@icgeb.org (G.R.); giulia.canarutto@icgeb.org (G.C.); 2Institute for Genetic and Biomedical Research, The National Research Council (CNR), 09042 Cagliari, Italy; michela.simbula@cnr.it; 3Department of Life Sciences, University of Trieste, 34127 Trieste, Italy; 4Dino Ferrari Center, Department of Pathophysiology and Transplantation, University of Milan, 20122 Milan, Italy; linda.ottoboni@googlemail.com (L.O.); stefania.corti@unimi.it (S.C.); 5Neurology Unit, Fondazione IRCCS Ca’ Granda Ospedale Maggiore Policlinico, 20122 Milan, Italy; 6Neuromuscular and Rare Diseases Unit, Fondazione IRCCS Ca’ Granda Ospedale Maggiore Policlinico, 20122 Milan, Italy; 7Department of Life and Environmental Sciences, University of Cagliari, 09042 Monserrato, Italy

**Keywords:** TDP-43, Rab4, MAP1B, ALS, FTD, endosomal trafficking, synaptic vesicle, NMJs, *Drosophila*

## Abstract

The pathological loss of nuclear TDP-43 is a hallmark of amyotrophic lateral sclerosis (ALS) and frontotemporal dementia (FTD), leading to extensive alterations in RNA metabolism and a broad number of neuronal transcripts. However, the key effectors linking TDP-43 dysfunction to synaptic defects remain unclear. In this study, using *Drosophila* and human iPSC-derived motoneurons, we identify Rab4 as a direct and conserved target of TDP-43, whose expression is necessary and sufficient to recover synaptic vesicle recycling, neuromuscular junction growth, and locomotor function in TDP-43-deficient motoneurons. Moreover, Rab4 activity promotes the presynaptic recruitment of *futsch*/MAP1B, a microtubule-associated protein also regulated by TDP-43, which autonomously supports synaptic growth and vesicle turnover. Together, these findings define a TDP-43/Rab4/*futsch*/MAP1B regulatory axis that couples endosomal dynamics to cytoskeletal assembly. Furthermore, this functionally coherent module provides a mechanistic basis for understanding how synaptic vulnerability is amplified in disease and offers a framework to identify key compensatory targets capable of sustaining neuronal function in the absence of TDP-43.

## 1. Introduction

Amyotrophic Lateral Sclerosis (ALS) is a progressive and fatal neurodegenerative disease characterized by the selective degeneration of upper and lower motor neurons in the brain and spinal cord [1,2,3,4]. This loss leads to progressive muscle weakness, paralysis, and ultimately respiratory failure, with most patients dying within 3–5 years of diagnosis [5,6]. A pathological hallmark of ALS is the cytoplasmic mislocalization and aggregation of the RNA-binding TDP-43 (TAR DNA-binding protein 43), observed in approximately 97% of sporadic and the majority of familial cases [7,8,9]. Normally localized to the nucleus, TDP-43 plays a central role in RNA metabolism, including transcriptional regulation, alternative splicing, RNA transport, and stability [10,11]. Its nuclear depletion and cytoplasmic aggregation disrupt these processes, leading to widespread transcriptomic dysregulation and the production of aberrant protein isoforms that impair neuronal homeostasis [12,13,14,15,16]. High-throughput studies, such as HITS-CLIP, have revealed that TDP-43 interacts with over 6000 RNA targets, nearly 30% of the neuronal transcriptome, underscoring its extensive regulatory capacity [17,18,19,20,21,22]. However, the scale of these interactions presents a major challenge: identifying the functionally relevant targets whose dysregulation contributes directly to motoneuron dysfunction and ALS pathogenesis. Moreover, establishing a functional hierarchy among TDP-43 targets, distinguishing core, pleiotropic effectors from more specialized or redundant transcripts, is essential for understanding molecular mechanisms underlying TDP-43 proteinopathies. In this context, it becomes increasingly important to identify metabolic and cellular pathways capable of restoring synaptic activity when TDP-43 function is compromised. With this idea in mind, in this study we combine *Drosophila* genetics with human iPSC-derived motoneurons to identify conserved molecular mechanisms capable of restoring synaptic function and neuromuscular innervation in TDP-43-deficient flies. By prioritizing conserved and functionally relevant targets, we aim to uncover actionable molecular pathways to deliver therapeutic strategies in ALS.

## 2. Results

### 2.1. TDP-43 Controls Synaptic Vesicle Recycling at NMJs

In previous work from our laboratory, we showed that the earliest consequences of TDP-43 loss of function are synaptic dysfunction and muscle denervation, driven by the downregulation of distinct synaptic proteins [23]. Intriguingly, we found that the mRNAs of some of these proteins are direct targets of TDP-43 (e.g., Syntaxin, DLG, *futsch*/MAP1B) [23,24,25,26,27]. In contrast, others, such as Synapsin and CSP, are downregulated despite lacking direct mRNA-TDP-43 interaction. This suggests that TDP-43 may regulate synaptic protein stability at the neuromuscular junction (NMJ) through alternative, RNA-dependent, and RNA-independent mechanisms. To investigate this further, we analyzed the behavior of synaptic recycling endosomes, key structures that support synaptic activity in neurons and motoneurons. For these experiments, we performed the FM1-43 dye uptake and unloading assays at the neuromuscular junctions (NMJs) of third instar *Drosophila* larvae (Appendix A). FM1-43 is a styryl dye that integrates into recycling synaptic vesicles during stimulation-induced endocytosis, allowing visualization of vesicle dynamics in live tissues [28]. NMJs were stimulated with high-potassium solution (90 mM KCl plus 1.5 mM calcium) during dye loading to evoke synaptic vesicle exocytosis and subsequent compensatory endocytosis. Dye unloading was triggered by a second round of depolarization, and residual fluorescence was quantified to assess vesicle release (Appendix A). Thus, we found that in wild-type larvae (w^1118^) FM1-43 loading resulted in a strong, well-circumscribed signal at the periphery of synaptic boutons, forming characteristic bright-shaped structures that reflect active vesicle recycling (Figure 1A). After unloading, the fluorescence intensity dropped significantly, confirming effective exocytosis and vesicle clearance. By contrast, NMJs from TDP-43–null mutants (*tbph*^∆23/∆23^) showed significantly altered FM1-43 staining. The signal was markedly reduced during loading (mean fluorescence intensity: 66.0 ± 3.6 a.u. in mutants vs. 100 ± 5.2 a.u. in controls; *p* < 0.0001, one-way ANOVA), and bouton morphology appeared irregular and disorganized (Figure 1B,C). Unloading efficiency was also severely impaired, with delta FM1-43 intensity (it is the delta intensity of loading signal “violet column” minus unload signal “gray column”) significantly reduced (19 ± 4.1% of delta intensity in mutants vs. 48 ± 4.0% in controls; *p* < 0.0001), suggesting defects in both endo- and exocytosis processes. To confirm that these phenotypes were specifically due to loss of TDP-43, we performed rescue experiments by expressing UAS-TBPH in the *tbph* mutant background using the pan-neuronal *elav*-GAL4 driver. In rescued larvae (*elav*-GAL4>UAS-TBPH, *tbph*^∆23/∆23^), both FM1-43 loading and unloading were significantly restored (loading: 82.0 ± 4.4 a.u.; unloading: 30.0 ± 3.4% of initial intensity; both *p* < 0.0001 compared to *tbph*^∆23/∆23^), and bouton morphology closely resembled that of controls (Figure 1A–C). This data supports the idea that TDP-43 activity is required to maintain presynaptic vesicle cycling, potentially via regulation of trafficking components, in vivo.

### 2.2. Rab4 Restores Locomotory Behaviors and Synaptic Growth in TDP-43-Null Flies

The marked impairment in synaptic vesicle recycling observed in TDP-43-null neuromuscular junctions (NMJs), as revealed by FM1-43 load/unload experiments, raised the question of what molecular effectors might mediate these defects. Given the established roles of Rab GTPases in regulating vesicular trafficking, we focused our attention on Rab4 and Rab5, key regulators of early and recycling endosomes [29]. Rab4 and Rab5 are small GTPases that control distinct aspects of vesicle sorting. Rab5 is primarily involved in the early endosome pathway, mediating the fusion and maturation of endocytic vesicles, including the transit of these cargoes to late endosomal compartments. In contrast, Rab4 regulates the rapid recycling of vesicles from the early endosomes back to the plasma membrane, a pathway essential for maintaining vesicle availability during sustained synaptic activity. These functions are particularly relevant at presynaptic terminals, where efficient vesicle turnover is crucial for neurotransmission [30,31,32,33,34,35,36,37,38]. Given the roles of these proteins, we hypothesized that the synaptic defects observed in the absence of TDP-43 might result from dysregulation of Rab4 or Rab5-mediated pathways. To test this hypothesis, we analyzed whether enhancing the function of these proteins could compensate for the NMJ defects caused by TDP-43 loss. For these experiments, we used the pan-neuronal elav-GAL4 driver to overexpress Rab4 or Rab5 in the TDP-43-null background (*elav*-GAL4>UAS-Rab4 or Rab5, *tbph*^∆23/∆23^). Morphological analysis of the larval NMJs showed a significant reduction in the number of synaptic branches in TDP-43 mutants compared to controls (*p* < 0.0001), confirming the role of this protein in synaptic development.

Remarkably, we observed that Rab4 expression led to a significant rescue of NMJ growth, restoring branch number and complexity to near wild-type levels (*p* < 0.001 vs. *tbph*^∆23/∆23^). In contrast, Rab5 overexpression failed to promote any substantial neuronal effect, supporting the specific requirement of Rab4 in maintaining synaptic growth downstream of TDP-43 function (Figure 2A,B).

To assess whether Rab4-mediated synaptic recovery translates into functional improvement in vivo, we evaluated the role of this protein in the locomotor behaviors of third instar *Drosophila* larvae. This behavioral readout integrates synaptic transmission and muscle innervation, constituting a sensitive indicator of neuromuscular performance. As expected, larvae lacking TDP-43 (to *elav*-GAL4>UAS-GFP, *tbph*^∆23/∆23^) exhibited pronounced locomotor deficits, characterized by reduced crawling speed and impaired coordination, reflecting the cumulative impact of presynaptic dysfunction and compromised muscle connectivity (Figure 2C,D).

Remarkably, the neuronal overexpression of Rab4 in the TDP-43-deficient background (*elav*-GAL4>UAS-Rab4, *tbph*^∆23/∆23^) significantly rescued larval locomotion compared to GFP-expressing mutants (*p* < 0.0001 compared to *elav*-GAL4>UAS-GFP, *tbph*^∆23/∆23^, Figure 2C), indicating that Rab4 overexpression is sufficient to restore not only the molecular and cellular features of the synaptic function, but also the neuromotor circuitry control in vivo.

In contrast, Rab5 overexpression failed to improve locomotor performance in identical TDP-43-deprived larvae (*elav*-GAL4>UAS-Rab5, *tbph*^∆23/∆23^), further supporting the specificity of Rab4 in rescuing the functional consequences of TDP-43 loss. Similar results were obtained using a motor neuron-specific driver, *D42*-GAL4 (Figure 2D).

### 2.3. Rab4 Promotes the Maturation of Presynaptic and Postsynaptic Terminals

To investigate the molecular basis of Rab4-mediated synaptic recovery, we found that Rab4 expression promotes the relocalization of the microtubule-binding protein *futsch*/MAP1B to presynaptic terminals (Figure 3A,B). This finding is particularly relevant, as *futsch* supports the growth and stabilization of motoneuron terminals by maintaining synaptic microtubule architecture during bouton addition and expansion. Strikingly, we observed that the presynaptic expression of Rab4 also restored the clustering of glutamate receptors (GluRIIA) and the scaffolding protein Disc large (DLG) at postsynaptic membranes. Accordingly, in TDP-43-deficient larvae, GluRIIA and DLG proteins appeared significantly reduced and disorganized, consistent with impaired presynaptic input. However, neuronal Rab4 overexpression restored their intensity and characteristic punctate distribution at the NMJ (Figure 3C–F), suggesting that Rab4 reactivation in presynaptic terminals is sufficient to enhance vesicle recycling and neurotransmitter release, thereby reestablishing effective communication with the postsynaptic muscle fiber. In contrast, Rab5 overexpression failed to rescue either presynaptic or postsynaptic markers, further supporting the specificity of Rab4 in compensating for synaptic defects caused by TDP-43 loss of function (Figure 3A–F).

### 2.4. TDP-43 Binds Rab4 mRNA and Regulates Its Expression Levels in Drosophila and Human Motoneurons

The results described above suggest that Rab4 may act downstream of, or in parallel to, TDP-43-regulated molecular pathways involved in supporting synaptic vesicle recycling and muscle innervation. To address these hypotheses, we analyzed the expression levels of *Rab4* mRNA in TDP-43 null brains by RT-qPCR and detected that the transcript levels of *Rab4* were significantly reduced in both *tbph*^∆23/∆23^ and *tbph* ^∆142/∆142^ alleles compared to wild-type controls (Figure 4A). Given that TDP-43 is an RNA-binding protein known to regulate mRNA stability, splicing, and transport, we next examined whether *Rab4* mRNA is a direct target of TDP-43 in vivo. For this, we performed RNA immunoprecipitation (RIP) experiments from *Drosophila* adult heads using flies expressing either the wild-type TDP-43 protein (functional RNA-binding protein) or a mutant version of TDP-43 carrying two-point mutations in the first functional RNA-binding domain [39]. The TDP-43-bound mRNAs were isolated and analyzed by RT-qPCR to detect a significant enrichment of *Rab4* mRNA in *Drosophila* TDP-43 immunoprecipitates compared with similar RIP experiments performed using the RNA-binding-defective mutant (Figure 4B), confirming that TDP-43 physically interacts with Rab4 mRNA in vivo in a manner that depends on its RNA-binding domain. Together, the data support a model in which TDP-43 stabilizes or regulates *Rab4* mRNA through direct binding, likely influencing its transcription or steady-state levels in vivo. In agreement with these results, we observed that *Rab4* transcript levels were similarly reduced in iPSC-derived human motoneurons lacking functional TDP-43 (Figure 4C), supporting the notion that these regulatory mechanisms would remain conserved during evolution, increasing the translational relevance of this regulatory axis.

### 2.5. Futsch/MAP1B Expression Recovers Presynaptic Vesicle Recycling, Synaptic Growth, and Locomotor Behavior in TDP-43 Mutant Flies

In the experiments above, we identified Rab4 as a direct target of TDP-43 and a remarkable regulator of synaptic activity, muscle innervation, and locomotor behaviors. Notably, Rab4 promotes the relocalization of *futsch*/MAP1B, a microtubule-associated protein critical for synaptic formation and stability, whose mRNA is also a direct target of TDP-43 [24]. These findings suggest that TDP-43 may coordinate the assembly of self-organizing protein circuits, with Rab4 and *futsch*/MAP1B acting together to support neuromuscular junction assembly and maintenance. To test whether Futsch/MAP1B expression alone is sufficient to compensate for TDP-43 loss, we restored its expression in TDP-43-null (*tbph*^∆23^) *Drosophila* using the pan-neuronal driver elav-GAL4. We first assessed whether Futsch could rescue synaptic vesicle cycling by performing FM1-43 dye uptake assays. In *tbph*^∆23^ mutants expressing UAS-*futsch* (*elav*-GAL4 > UAS-*futsch*; *tbph*^∆23^/*tbph*^∆23^), FM1-43 labeling in presynaptic boutons regained the characteristic ring-shaped pattern, comparable to wild-type and in sharp contrast to the diffuse and disorganized staining observed in controls expressing GFP (*elav*-GAL4 > UAS-GFP; *tbph*^∆23^/*tbph*^∆23^). This indicates that neuronal expression of this microtubule-binding protein is sufficient to recover presynaptic vesicle recycling capacity in the absence of TDP-43 (Figure 5A–C). Next, we examined NMJ morphology and found that *futsch* expression resulted in a significant increase in the number of synaptic boutons, enhanced terminal branching, and greater axonal arbor complexity, expanding the muscle innervation area compared to GFP-expressing controls (Figure 5D,E). Finally, we observed that the larvae expressing *futsch* exhibited a striking increase in locomotor activity, as measured by larval crawling assays, compared to TDP-43-null controls expressing GFP (Figure 5F), indicating that the described structural and functional synaptic improvements effectively translated into a rescue of locomotor behavior.

## 3. Discussion

### 3.1. TDP-43 Dysfunction and the Challenge of Transcriptomic Complexity in ALS

TDP-43, a key factor in ALS pathogenesis, plays a central role in RNA metabolism, and its altered function results in the dysregulation of thousands of RNA targets, complicating the efforts to define which downstream effectors are functionally relevant [40]. Therefore, it remains unclear whether the onset and progression of the disease reflect the accumulation of multiple small hits across a disorganized transcriptomic landscape or if a rather coherent pathophysiological mechanism exists. In this study, we demonstrate that the absence of TDP-43 disrupts synaptic vesicle recycling at the neuromuscular junction (NMJ) in *Drosophila* larvae, leading to reduced endo and exocytosis efficiency, abnormal bouton morphology, and impaired synaptic connectivity. These findings are consistent with the idea that synaptic vulnerability is a primary and conserved consequence of TDP-43 dysfunction, even preceding neuronal loss [41,42].

### 3.2. Rab4-Dependent Vesicle Recycling Arises as a Functional Bottleneck Regulated by TDP-43

Several groups have identified vesicular transport defects in ALS, implicating Rab proteins, retromer components, and endosomal–lysosomal pathways [43]. Yet the mechanistic links with TDP-43 have remained elusive. Our identification of Rab4 as a direct TDP-43 target fills this gap. Rab4, a small GTPase involved in fast recycling from early endosomes, was sufficient to restore vesicle recycling, synaptic terminals growth, and locomotive behaviors in TDP-43-null backgrounds but not in wild-type, indicating a specific gene dose-dependent compensatory effect rather than a general enhancement of synaptic development (Appendix A).

This specificity is particularly striking, as Rab5, a related GTPase involved in early endosomal trafficking, fails to rescue these defects, suggesting a preferential requirement for the Rab4-mediated pathway. Our findings reinforce recent reports highlighting Rab proteins roles in synaptic vesicle mobilization, trafficking and receptor recycling in vitro using mammalian neurons [44] and support the view that impairments in these functions may represent early pathogenic events contributing to synaptic dysfunction in ALS.

### 3.3. MAP1B/futsch Links Vesicle Dynamics and Cytoskeletal Assembly Through a Self-Reinforcing Module

While vesicle trafficking is essential for synaptic transmission, presynaptic architecture depends on dynamic cytoskeletal remodeling. We demonstrate that *futsch*, the *Drosophila* homolog of MAP1B and a direct target of TDP-43, is also sufficient to rescue multiple aspects of synaptic function in TDP-43 null motoneurons. Thus, we found that *futsch* not only rescued bouton morphology and NMJ growth, but also significantly improved vesicle recycling efficiency and locomotor performance, revealing an unexpected role in sustaining presynaptic vesicle dynamics beyond its established function in microtubule stabilization. Mechanistically, we propose that *futsch* supports vesicle release by coupling microtubule stability with the organization of the submembrane cytoskeleton, thereby maintaining efficient vesicle recycling near active zones. Although microtubules do not normally extend deeply into presynaptic boutons, short segments and associated proteins such as MAP1B and Tau can interface with actin and spectrin networks to sustain vesicle pool organization and transport. Within this framework, *futsch* likely serves as a dynamic bridge between microtubules and the cortical actin network, coordinating cytoskeletal remodeling with vesicle mobilization to preserve synaptic efficacy [45,46,47,48,49,50,51]. Interestingly, Rab4 overexpression also promoted *futsch* recruitment to presynaptic terminals, suggesting that Rab4-mediated vesicle recycling and cytoskeletal remodeling constitute an integrated and mutually reinforcing module (Figure 6). This loop may represent a critical vulnerability point in ALS pathogenesis and a promising target for future therapeutic strategies aimed at stabilizing synaptic integrity in neurodegenerative disease.

In healthy motor neurons (left), wild-type TDP-43 regulates the expression of both *futsch* and *Rab4* via direct mRNA interactions. This supports microtubule organization through MAP1B/*futsch* stabilization and maintains proper vesicle trafficking. Rab4-positive vesicles from the releasable pool are efficiently mobilized to the presynaptic membrane, ensuring neurotransmitter release, while Rab5 regulates endosomal recycling and the lysosomal targeting. Futsch further stabilizes the microtubule network and promotes bouton growth, which is essential for the organization and anchoring of vesicular components. Together, this creates a functional feedback loop: Rab4 enables vesicle recycling, while *futsch* reinforces synaptic architecture and supports vesicle pool maintenance, ensuring sustained neurotransmission. In contrast, in affected motor neurons (right), TDP-43 dysfunction—characterized by pathological mislocalization and loss of RNA-binding activity—compromises cytoskeletal stability and disrupts Rab4-dependent vesicle mobilization, breaking this feedback loop. These defects collectively impair synaptic transmission and contribute to motor neuron degeneration observed in ALS and related TDP-43 proteinopathies. Created with (BioRender.com).

### 3.4. A Modular Response Capable of Bypassing TDP-43 Loss of Function

Perhaps the most compelling aspect of our findings is that Rab4 and *futsch*, direct targets of TDP-43, are sufficient to restore synaptic function, morphology, and behavior in the complete absence of TDP-43. This supports a model in which TDP-43 acts not simply as a global RNA chaperone, but as a regulator of discrete, high-impact modules. This model is conceptually aligned with recent transcriptomic and proteomic analyses that reveal discrete “regulons” downstream of other ribonuclear proteins (RBPs) like FUS, TIA1, including TDP-43 [52]. The data also aligns with the increasing recognition of compartmentalized RNA transport and metabolism at synaptic sites and the requirements for local translation to sustain plasticity and repair.

### 3.5. Implications for ALS and Neurodegenerative Disease

Our findings provide several advances to the current ALS field. First, they demonstrate that Rab4 and MAP1B/*futsch* represent conserved, tractable nodes of vulnerability in TDP-43 proteinopathies. Their ability to restore synaptic function offers proof-of-concept for therapeutic strategies targeting a small subset of selected TDP-43-regulated genes, rather than the entire transcriptomic cascade. Second, they integrate two central features of early neurodegeneration, vesicular trafficking and cytoskeletal breakdown, into a unified mechanism, addressing the long-standing question of how synaptic and structural defects interact in ALS. Finally, the conservation of Rab4 regulation across species underscores the translational value of *Drosophila* models and strengthens the rationale for moving toward mammalian validation.

## 4. Material and Methods

### 4.1. Fly Strains

The complete genotypes of the fly stocks are indicated below:

*w*^1118^-w; *tbph*^Δ23^/CyOGFP [53]-w; *tbph*^Δ142^/CyOGFP-w; *elav*-GAL4-w; *D42*-GAL4-w; UAS-TBPH-w; UAS-TBPH^mutF/L^-w; UAS-Rab4-GFP-w; UAS-Rab5-GFP-yw; UAS-mCD8::GFP-P{EP}*futsch*^EP1419^ w^1118^ (BDSC#10751).

### 4.2. Fly Maintenance

All flies’ stocks were maintained on a standard cornmeal medium (agar 6 g/L, 62.5 g/L yeast, 41.6 g/L sugar, 29 g/L flour, propionic acid 4.1 ml/L) at 25 °C under a 12 h light/dark cycle and constant humidity.

### 4.3. Larval Movement

Following a previously established protocol [53], we assessed the peristaltic waves of third instar larvae. Larvae were first selected and rinsed with water to remove any food residue. They were then transferred to a Petri dish (94 × 16 mm) containing a layer of 0.7% agarose in distilled water. After a 30–60 s adaptation period, we analyzed their locomotion by counting the number of peristaltic waves over a 2 min interval.

### 4.4. Climbing Assay

One-day-old adult flies (equal numbers of males and females) were transferred to fresh food vials and maintained under standard conditions. On day 7, climbing ability was tested using a 50 mL glass cylinder divided into three 5 cm sections. Flies were gently tapped to the bottom, and the number reaching the top section (>10 cm) within 15 s was recorded after a 30 s adaptation period. Three independent trials were performed per vial, and the average score was calculated. At least 60 flies were analyzed per genotype.

### 4.5. NMJs Characterization

The anatomical organization of a third-instar *Drosophila* larva exhibits a clearly repeated segmental pattern, in which the muscle architecture is highly conserved. To ensure reproducibility and minimize variability related to anatomical differences, analyses were restricted to muscles 6 and 7 in the second abdominal segment. Synaptic branches refer to the arborizations of the motor neuron terminals at the neuromuscular junction (NMJ), whose reactivity with horseradish peroxidase (HRP) is well established and routinely used to label the presynaptic terminals of *Drosophila* and other insect neurons [54].

### 4.6. FM 1-43 In Vivo Assay and Quantification

The FM 1-43 assay protocol was performed as previously published [28,55,56,57]. The procedure was as follows: FM 1-43 4 mM dye uptake was induced by a 1 min stimulation with 90 mM KCl. The dye was then unloaded with a second 7 min stimulation of KCl. Images were captured using a Nikon DS-Qi2 microscope equipped with a 60×/1.0 W objective lens. Bouton intensities were quantified using ImageJ software (v.1.54). The percentage of the area occupied by the FM 1-43 dye within each synaptic bouton was calculated by manually outlining the bouton’s perimeter. All results were normalized to the *w*^1118^ control after the loading phase. FM1-43 imaging, as well as all other confocal imaging analyses, were quantified in a blinded manner. The operator acquired all images and assigned anonymized codes, ensuring that the person performing the quantification was blind to the experimental conditions.

### 4.7. Immunohistochemistry

Third instar larval body were dissected in saline solution (CaCl_2_ 0.1 mM, MgCl_2_ 4 mM, KCl 2 mM, NaCl 128 mM, sucrose 35.5 mM and Hepes 5 mM pH 7.2) [58]. The tissue was fixed for 20 min in 4% paraformaldehyde, with exception of samples stained for anti-GluRIIA, which were fixed for 5 min in methanol at −20 °C. Following fixation, samples were washed in PBS 0.1% Tween20 and blocked for 1 h with 5% Normal Goat Serum (Vector Laboratories, Newark, CA, USA) in PBS 0.1% Tween20. Primary antibodies were incubated over night at 4 °C, and secondary antibodies were incubated for 2 h at room temperature. Samples were mounted in SlowFade Gold (Life Technologies, Carlsbad, CA, USA). Images were acquired on a Zeiss 880 Airyscan using 40× and 63× oil objectives and then analyzed using ImageJ (Wayne Rasband, NIH, Bethesda, USA). Antibody dilutions were as follows: anti-HRP (Jackson 1:150), anti-Dlg 4F3c (DSHB 1:250), anti-GluRIIA 8B4D2 (DSHB 1:15), anti-Futsch #22C10s (DSHB 1:50), anti-HRP-Cy3 (Jackson 1:150), Alexa-Fluor^®^ 488 (mouse, rabbit 1:500), Alexa-Fluor^®^ 555 (mouse, rabbit 1:500).

### 4.8. Quantification of Confocal Images

Animals were processed simultaneously, and all images were acquired using the same microscope settings to ensure consistency. The neuromuscular junctions (NMJs) on muscles 6 and 7 of the second abdominal segment were selected for analysis. Images were processed using ImageJ software, and statistical analysis was performed with Prism (GraphPad, San Diego, CA, USA). For the quantification of pre- and postsynaptic markers, samples were double labeled with anti-HRP and the marker of interest. The mean intensity of both markers was quantified, and a ratio of marker intensity to HRP intensity was calculated [23,59].

### 4.9. RNA Extraction

Total RNA was extracted from dissected larval brains or whole adult heads or iPSC-derived motoneuron using TRIzol reagent (Cat. #15596026, Invitrogen, Waltham, Massachusetts, USA) according to the manufacturer’s instructions. RNA was DNAse treated with TURBO DNA-*free*™ Kit (#AM1907, Thermo Fisher Scientific, Waltham, Massachusetts, USA). First-strand cDNA was synthesized with the SuperScript VILO Master Mix (Thermo Fisher) or SuperScript III First-Strand Synthesis System (Cat. #18080-051, Invitrogen).

### 4.10. qRT-PCR

Real time PCR was carried out using Platinum SYBR Green (#11744-100, Invitrogen) on a Bio-Rad CFX96 qPCR System. Below in Table 1 the used primers:

### 4.11. Immunoprecipitation for RNA Enrichment

Flash-frozen *Drosophila* heads from elav-GAL4/UAS-TBPH and elav-GAL4/+;UAS-TBPH^mutF/L^/+ genotypes were homogenized in immunoprecipitation buffer (20 mM Hepes, 150 mM NaCl, 0.5 mM EDTA, 10% glycerol, 0.1% Triton X-100, and 1 mM DTT supplemented with protease inhibitors (Roche #11836170001)) using a Dounce homogenizer as described [60]. The lysate was then subjected to 0.4× g centrifugation for 5 min to remove debris. Protein G magnetic beads (#10003D, Thermo Fisher Scientific) coated with anti-FLAG-M2 antibody (#F3165, Sigma, Burlington, Massachusetts, USA) were added to the cleared lysates and incubated for 30 min at 4 °C. After five washes with immunoprecipitation buffer, RNA was extracted from the beads using TRIzol reagent.

### 4.12. Statistical Analysis

Statistical analysis was performed using Prism (GraphPad, San Diego, CA, USA) version 10.2.3. Statistical tests included one-way ANOVA with a Bonferroni correction and two-tailed *t*-tests. All values in figures are presented as the mean ± standard error of the mean (SEM). Statistical significance is indicated by * *p* < 0.05, ** *p* < 0.01, *** *p* < 0.001 and **** *p* < 0.0001.

## Figures and Tables

**Figure 1 ijms-26-11030-f001:**
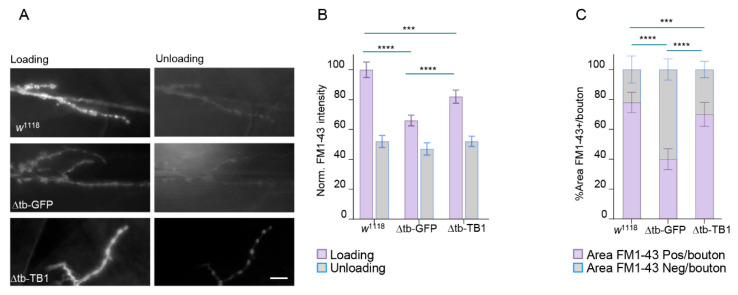
TDP-43 regulates synaptic vesicle recycling at NMJs. (**A**) In vivo image of third instar NMJ terminals of muscle 6/7 s segment stained with FM1-43 dye to show the vesicle upload and unload after chemical stimulation in *w*^1118^–∆tb-GFP (*tbph*^∆23^, *elav*-GAL4/*tbph*^∆23^, UAS-GFP) and ∆tb-TB (*tbph*^∆23^, *elav*-GAL4/*tbph*^∆23^, UAS-TBPH) genotypes. (**B**) Quantification of FM1-43 intensity after the loading (violet column) and then the unloading (gray column) of the synaptic vesicles of the panel A. (**C**) Quantification of percentage of the FM1-43 occupied area per single boutons. *n* ≥ 15 larvae; *n* > 100 boutons. *** *p* < 0.001, **** *p* < 0.0001, calculated by one-way ANOVA, error bars SEM. Scale bar 40 µm.

**Figure 2 ijms-26-11030-f002:**
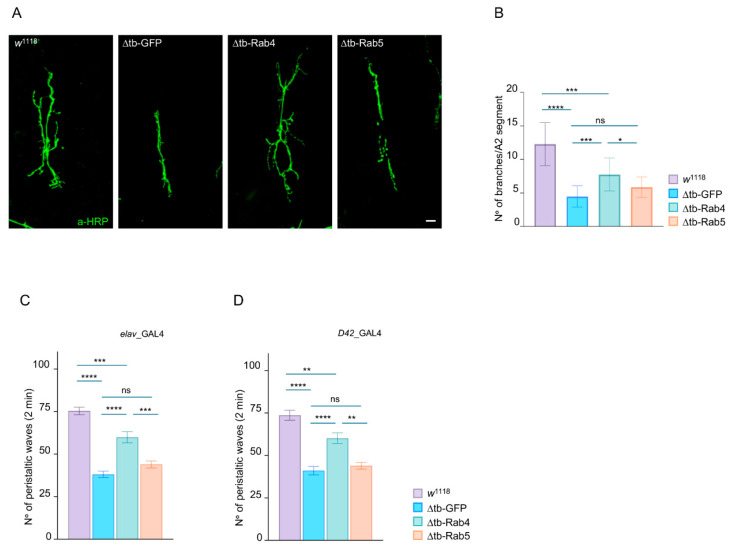
Rab4 rescues synaptic growth and function in TDP-43-null NMJs. (**A**) Confocal image of third instar NMJ terminals in muscle 6/7 s segment stained with anti-HRP (in green) in *w*^1118^, ∆tb-GFP, ∆tb-Rab4 and ∆tb-Rab5. (**B**) Quantification of branches number. *n* ≥ 15. (**C**,**D**) Number of peristaltic larval waves of third instar larvae during 2 min of *w*^1118^, ∆tb-GFP, ∆tb-Rab4 and ∆tb-Rab5 using *elav*-GAL4 driver (**C**) or *D42*-GAL4 driver (**D**). Genotype: *w*^1118^, ∆tb-GFP (*tbph*^∆23^, *elav*-GAL4/*tbph*^∆23^, UAS-GFP or *tbph*^∆23^/*tbph*^∆23^, UAS-GFP; *D42*-GAL4/+), ∆tb-Rab4 (*tbph*^∆23^, *elav*-GAL4/*tbph*^∆23^; UAS-Rab4-GFP/+ or *tbph*^∆23^/*tbph*^∆23^; UAS-Rab4-GFP/*D42*-GAL4) and ∆tb-Rab5 (*tbph*^∆23^, elav-GAL4/tbph^∆23^; UAS-Rab5-GFP/+ or *tbph*^∆23^/*tbph*^∆23^; UAS-Rab5-GFP/*D42*-GAL4). *n* ≥ 15 larvae, ns = not significant, * *p* < 0.05, ** *p* < 0.01, *** *p* < 0.001, **** *p* < 0.0001, calculated by one-way ANOVA, error bars SEM. Scale bar 20 µm.

**Figure 3 ijms-26-11030-f003:**
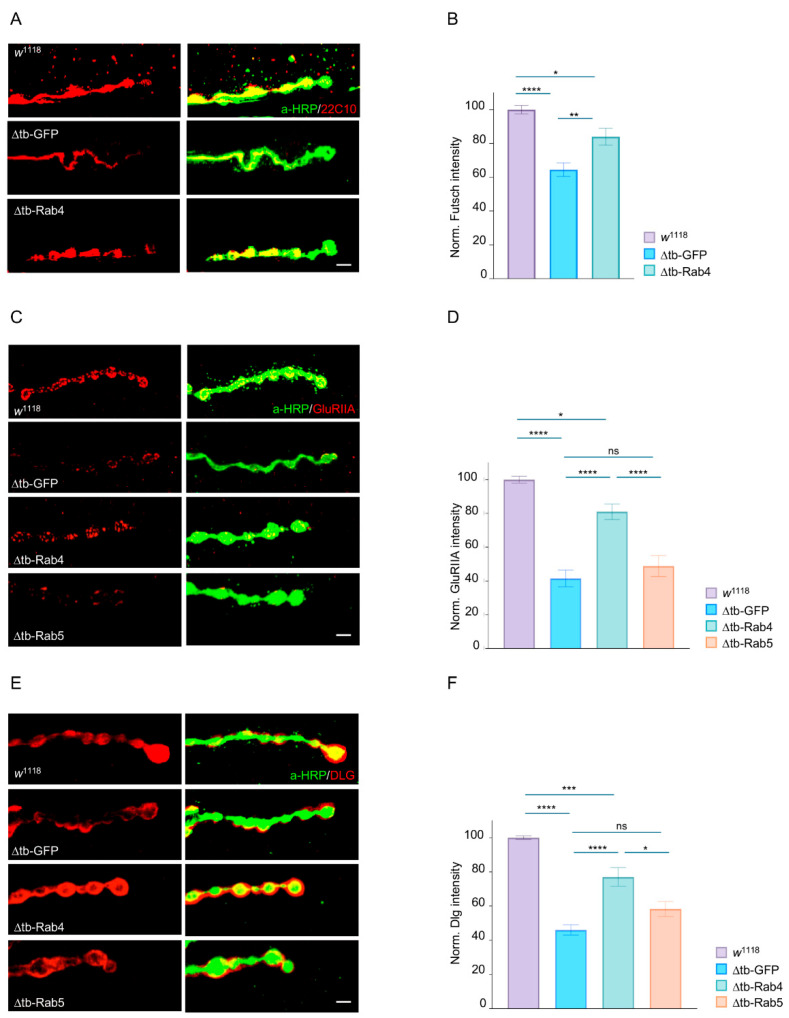
Rab4 promotes the assembly of pre- and postsynaptic compartments. (**A**) Confocal image of third instar NMJ terminals of muscle 6/7 s segment in *w*^1118^, ∆tb-GFP and ∆tb-Rab4 stained with anti-HRP (in red) and Futsch (22C10 in fuchsia) with relative quantification in (**B**). (**C**) Confocal image of third instar NMJ terminals of muscle 6/7 s segment in *w*^1118^, ∆tb-GFP, ∆tb-Rab4 and ∆tb-Rab5 stained with anti-HRP anti-HRP (in green) and Glutamate receptor IIA (GluRIIA in red) with relative quantification in (**D**). In panel (**E**) same genotypes stained with anti-HRP (in green) and Disc large (DLG in red) with relative quantification in (**F**). Genotype: *w*^1118^, ∆tb-GFP (*tbph*^∆23^, *elav*-GAL4/*tbph*^∆23^, UAS-GFP), ∆tb-Rab4 (*tbph*^∆23^, *elav*-GAL4/*tbph*^∆23^; UAS-Rab4-GFP/+) and ∆tb-Rab5 (*tbph*^∆23^, *elav*-GAL4/*tbph*^∆23^; UAS-Rab5-GFP/+). All intensity signals were normalized to the HRP signal, ensuring independence from terminal size. *N* ≥ 15 larvae; *n* > 200 boutons, ns = not significant, * *p* < 0.05, ** *p* < 0.01, *** *p* < 0.001, **** *p* < 0.0001, calculated by one-way ANOVA, error bars SEM. Scale bar 5 µm.

**Figure 4 ijms-26-11030-f004:**
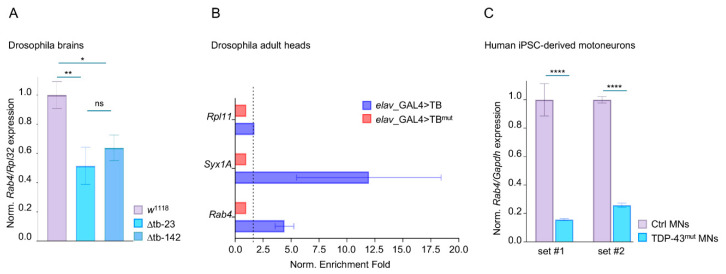
TDP-43 directly binds *Rab4* mRNA and regulates its expression levels in *Drosophila* and human iPSC-derived motoneurons. (**A**) qRT-PCR of *Rab4* transcript levels normalized on *Rpl32* (housekeeping) in third instar larval brains of w1118, ∆tb-23 and ∆tb-142 (*tbph*^∆23^/*tbph*^∆23^ and *tbph*^∆142^/*tbph*^∆142^) *n* ≥ 15 larval brains, ns = not significant, * *p* < 0.05, ** *p* < 0.01, calculated by one-way ANOVA, error bars SEM. (**B**) qRT-PCR analysis of mRNAs immunoprecipitated by FLAG-tagged TBPH (*elav*-GAL4>TB = *elav*-GAL4/UAS-TBPH) and its mutant variant *elav*-GAL4>TB^mut^ = *elav*-GAL4/+;UAS-TBPH^mutF/L^/+). *Rab4* enrichment folds were quantified by normalizing TBPH^mut^ to a value of 1, with *Rpl11* and *Syntaxin1A* serving as the negative and positive controls, respectively. *n* = 2 (biological replicates). (**C**) qRT-PCR analysis of *Rab4*. *Rab4* mRNA levels from human iPSC-derived motoneurons were normalized on *Gapdh* levels in two set1 of control sample (healthy individual) and ALS patients (TDP-43 mutated). *n* = 3, **** *p* < 0.0001, calculated by *t*-test, error bars SEM.

**Figure 5 ijms-26-11030-f005:**
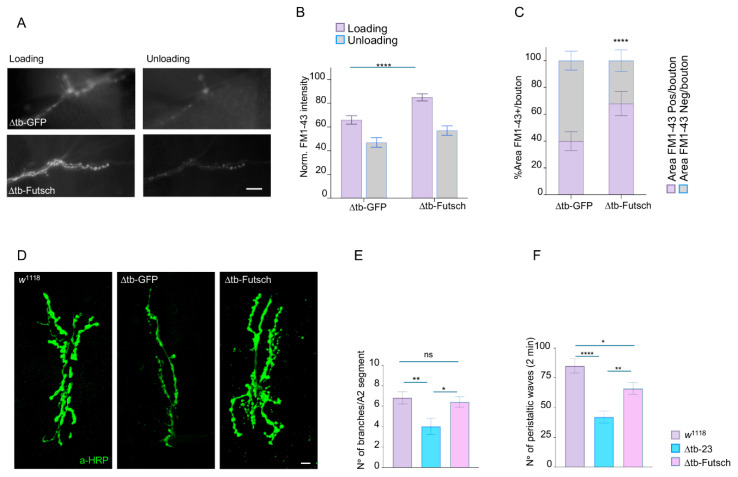
*Futsch*/MAP1B expression recovers presynaptic vesicle recycling, synaptic growth, and locomotor behavior in TDP-43 mutant flies. (**A**) In vivo image of third instar NMJ terminals of muscle 6/7 s segment stained with FM1-43 dye to show the vesicle upload and unload after chemical stimulation in ∆tb-GFP (*tbph*^∆23^,*elav*-GAL4/*tbph*^∆23^,UAS-GFP) and ∆tb-Futsch (UAS-EP10756/+; *tbph*^∆23^, *elav*-GAL4/*tbph*^∆23^) *n* ≥ 15 larvae; *n* > 100 boutons. (**B**) Quantification of FM1-43 intensity after the loading (violet column) and then the unloading (gray column) of the synaptic vesicles of the panel A. (**C**) Quantification of FM1-43 positive area percentage per single boutons. *n* > 100 boutons. (**D**) Confocal image of third instar NMJ terminals of muscle 6/7 s segment in *w*^1118^, ∆tb-GFP and ∆tb-Futsch stained with anti-HRP (in green) *n* ≥15 larvae. Relative quantification of branches number in (**E**). (**F**) Number of peristaltic larval waves of third instar larvae during 2 min of *w*^1118^, ∆tb-GFP, ∆tb-Futsch. *n* ≥ 15 larvae. ns = not significant, * *p* < 0.05, ** *p* < 0.01, **** *p* < 0.0001 calculated by one-way ANOVA, error bars SEM. Scale bar 40 µm (**A**) and 20 µm (**D**).

**Figure 6 ijms-26-11030-f006:**
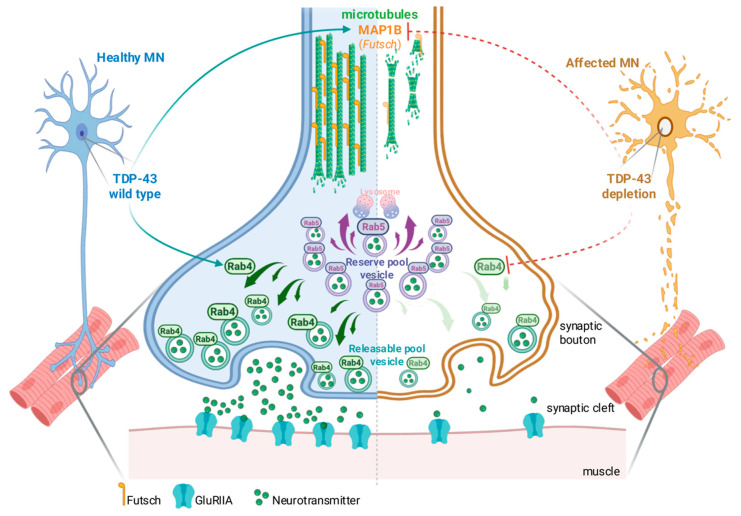
TDP-43 maintains NMJs connectivity via synergistic Rab4 and *futsch*/MAP1B pathways.

**Table 1 ijms-26-11030-t001:** List of primer sequences used in this work.

Target	Species	Forward	Reverse
***Rab4*** (Figure 4A)	*D. mel*	5′-CTGAATGATGCTCGCACTCT-3′	5′-AAGGTGCTTGCTTCCAGAAA-3′
***Rab4*** (Figure 4B)	*D. mel*	5′-GGCAGCGGTAAGAGTTGTCT-3′	5′-GAATCTCTCCTGACCGGCTG-3′
** *Syx1A* **	*D. mel*	5′-TGTTCACGCAGGGCATCATC-3′	5′-GCCGTCTGCACATAGTCCATAG-3′
** *Rpl11* **	*D. mel*	5′-CCATCGGTATCTATGGTCTGGA-3′	5′-CATCGTATTTCTGCTGGAACCA-3′
** *Rpl32* **	*D. mel*	5′-AAGCGGCGACGCACTCTGTT-3′	5′-GCCCAGCATACAGGCCCAAG-3′
** *Rab4* **	*human*	5′-CAGAAAGAATGGGCTCAGGT-3′	5′- TGCTCTCCTAACAACCACACT-3′
** *Gapdh* **	*human*	5′-GTCCACTGGCGTCTTCAC-3′	5′-AGGCATTGCTGATGATCTTGA-3′

## Data Availability

The raw data supporting the conclusions of this article will be made available by the authors on request.

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
