# Peer review of "TDP-43 Regulates Rab4 Levels to Support Synaptic Vesicle Recycling and Neuromuscular Connectivity in Drosophila and Human ALS Models"

_ijms, 2025, doi:10.3390/ijms262211030_

Round 1
Reviewer 1 Report
Comments and Suggestions for Authors
Gbadamosi et al. use a drosophila model of TDP43 loss to explore the consequence of Rab4 overexpression on synaptic vesicle recycling at the NMJ. They find that Tdp43 loss causes defective synaptic vesicle recycling, defects in cytoskeletal architecture, and structural and functional NMJ defects. They explore the role of two proteins known to be involved in vesicle recycling, namely, Rab4 and Rab5 and find that Rab4 overexpression rescues the NMJ defects and increases Map1B labelled axonal structures in the pre-synapse, and that Rab4 is a direct target of Tdp43 in drosophila embryos. Finally, they show that direct overexpression of Map1B can also rescue the vesicle loading and NMJ defects.
This is a well-designed study and the manuscript is also structured well. The role of TDP43 in vesicle recycling and NMJ defects is well established. The identification of Rab4 as a protein that can rescue some of these defects is new. Though its not clear why the authors chose Rab4 and Rab5 amongst the dozens of Rab genes involved in cargo trafficking.
There are some details and experiments missing in the manuscript. My comments are as below:
- Fig 1A: ∆tb-Futsch included in the legend but not in the figure
- 1B, C need to be referred in the text. Also, expression of TDP43 does not rescue the recycling defects completely. Significance calculations between wild type and recuse value should be reported, and this should be commented on. Line 88 should refer to Fig. 1B,C. Authors claim unloading efficiency is impaired. From the graphs, it seems that the loading process is affected with not much difference found in the unloading event. Representative images to show how area of FM1-43 per bouton was calculated are required for Fig.1
- “Morphological analysis of the larval NMJs showed a significant reduction in the number of synaptic branches in TDP-43 mutants compared to controls (p < 0.0001), confirming the role of this protein in synaptic development”. Page 4, line29. What do the authors mean by “synaptic branches”? Also, Fig 2A needs to be referred to here. Please explain what an A2 segment is. What is the alpha-HRP staining? Line 158 in the figure legend should say C and D instead of A and B.
- Were the MAP1B levels normalised to HRP area? What happens to total MAP1B levels after Rab4 overexpression? Is this relocalisation versus an increase in axonal length?
- The effect of Rab4 on vesicle recycling needs to be shown as this is a major claim in this study.
- What happens if rab4 is overexpressed in wild type embryos? Are the observations a generic effect of rab4 overexpression?
- Does TDP43 bind Rab4 in human neurons or tissue? TDP43 eCLIP data is publicly available from the ENCODE consortium or from Jernej Ule’s lab. It might be worth looking at these datasets, especially if these results are to be broadly applicable to ALS.
- A previous study found that TDP43 loss has not effect on RAB4 + endosomal trafficking in dendrites (PMID: 27621269 ). Could the authors discuss why their results differ?
Reviewer 2 Report
Comments and Suggestions for Authors
Relocation of TDP-43 from the nucleus to the cytoplasm is the hallmark of LAS and FTD. Under normal conditions, nuclear TDP-43 is a master regulator of RNA metabolism. While transcriptomic studies have identified numerous potential targets, the key effectors linking TDP-43 dysfunction to specific pathologic outcomes, such as synaptic failure, remain poorly understood. In this study, Gbadamosi et al. use Drosophila and human iPSC-derived motoneurons to identify the small GTPase Rab4 as a critical functional target. They discovered that TDP-43 directly regulates Rab4 mRNA, the loss of which results in defective synaptic vesicle recycling and impaired NMJ integrity. Strikingly, they demonstrate that neuronal overexpression of either Rab4 or the microtubule-associated protein Futsch is sufficient to rescue these synaptic and behavioral deficits in TDP-43 null models, proposing an integrated TDP-43/Rab4/Futsch axis essential for synaptic homeostasis.
- Fig 1B, there are inconsistencies between the text and figures. The text describes a significant difference in unloading (WT vs KO vs rescue = 48% vs 19% vs 30%), but the bar graphs for all genotypes appear to show values above 40%. Fig 1C, it is unclear whether the "FM1-43 occupied area" is quantified for the loading or unloading phase.
- The assay measures two distinct vesicle pools. The "loading" phase labels a large population of newly endocytosed vesicles, which are not immediately ready to release. The "unloading" phase primarily reflects the fusion of the pre-existing releasable pool. Therefore, the assay does not track a single, synchronous cohort of vesicles. The authors' interpretation of a direct defect in "unloading" in the mutant should be refined to acknowledge that the observed impairment likely reflects a failure to maintain or replenish the releasable pool, rather than a direct defect in exocytosis of primed vesicles.
- Rab3, but not Rab4, is the dominant Rab present on mature synaptic vesicles. So the rescue of synaptic vesicle unloading by Rab4 might be indirect, for example, via enhancing replenishment of releasable synaptic vesicles. Rab4 on mature synaptic vesicles in the graphic model should be revised.
- The finding that Futsch, a microtubule-associated protein, rescues vesicle unloading is surprising, as microtubules typically do not extend into the presynaptic bouton where vesicle fusion occurs. The authors should provide a more detailed mechanistic hypothesis to reconcile the apparent discrepancy.
- Given that TDP-43 regulates thousands of transcripts, the claim that overexpressing a single downstream target like Rab4 or Futsch is "sufficient" to fully restore synaptic function risks overinterpretation. The quantitative data themselves show partial, albeit significant, rescue, which is more biologically plausible. The authors should tone down their conclusions throughout the manuscript, acknowledging that while Rab4 and Futsch represent critical nodes within the TDP-43 regulatory network, they are likely part of a broader compensatory mechanism rather than singular master switches.
Reviewer 3 Report
Comments and Suggestions for Authors
The research topic is interesting and attractive to specialists in neurochemistry, neurophysiology, and neuropharmacology. Some comments are needed. Comments: 1. The introduction needs to be revised. It is difficult to understand why the authors chose these research topics. The introduction needs to be expanded to include material on Rab4 and its relationship with TDP43 in various types of CNS and cardiac pathologies, and oncology. 2. There is no study objective. 3. There is no study design. Fly and human models are used. 4. Move some material from the Results section to the Discussion section. 5. Add research perspectives in the Discussion section in the following areas: Rab4 as a promising target for the treatment of neurodegenerative pathologies and as a biomarker. Promising pharmacological agents may be mentioned. 6. There are no research limitations.
Reviewer 4 Report
Comments and Suggestions for Authors
The study is promising and potentially impactful, but it requires additional experimental validation, particularly at the protein level and in the human model, to support the proposed TDP-43/Rab4/Futsch regulatory axis. The Discussion implies that Rab4 and Futsch are sufficient to restore neuronal homeostasis in the absence of TDP-43. This is an overinterpretation given that only NMJ and locomotion phenotypes were examined. Broader neuronal or survival endpoints are not evaluated.
Major comments:
- While the authors demonstrate reduced Rab4 mRNA and TDP-43–dependent binding, they do not establish functional causality between TDP-43–mediated transcriptional control and Rab4 protein levels. Western blot or immunostaining data confirming Rab4 protein reduction in TDP-43 mutants are missing. The use of only qRT-PCR (without protein validation) weakens the claim that Rab4 is a “direct and conserved target.”
- The rescue of NMJ structure and behavior by Rab4 overexpression is intriguing but could reflect a generic enhancement of vesicle recycling rather than specific compensation for TDP-43 loss. It would be important to show whether Rab4 overexpression in a wild-type background produces no major gain-of-function phenotype (to rule out non-specific effects).
- The authors propose a TDP-43/Rab4/Futsch module but provide limited mechanistic data linking these components. Does Futsch overexpression affect Rab4 localization or activity?Conversely, does Rab4 overexpression alter Futsch expression levels? The relationship remains correlational rather than causal.
- Human data are limited to a single qPCR analysis from iPSC-derived motoneurons with TDP-43 mutations. No functional assays (e.g., vesicle recycling, neurite morphology, or electrophysiology) are shown in the human model. The claim of evolutionary conservation is therefore overstated.
- The FM1-43 dye assays are standard but largely qualitative. Quantitative results (number of boutons analyzed, replicate numbers, and raw values) are insufficiently detailed. The authors should clarify whether the imaging and quantification were blinded to genotype.
- In several figures (e.g., Fig. 2–5), sample sizes (n values) are given per bouton or per branch, which may not represent independent biological replicates. Statistical analysis should be performed on biological replicates (larvae or animals), not on subcellular structures.
Round 2
Reviewer 2 Report
Comments and Suggestions for Authors
The authors have addressed my concerns.
Reviewer 3 Report
Comments and Suggestions for Authors
Dear Authors! Thank you for your positive response and understanding! Your work is very interesting and has significant scientific significance. I support it. I don't want to seem too tedious or picky, but I offer my counter-comments. I've always believed that the research objective of any study should be clear and clearly stated. The limitations of the study and its future prospects are separate sections of the article. I would also like to know your opinion on the prospects and feasibility of pharmacological interventions against Rab4 as a direct and conserved target of TDP-43.I wish you success in your work and scientific achievements.